# DeMo: Decoupled Momentum Optimization

## ABSTRACT

Training large scale neural networks typically involves sharing the gradients between all accelerators, which necessitates specialized high-speed interconnects. Taking cues from signal processing, we show that it is not necessary to share or synchronize the full optimizer states and model parameters during training. By decoupling the momentum and allowing divergence in the optimizer states across accelerators, it is possible to even improve convergence compared to previous state of the art optimizers. From this, we introduce **De**coupled **Mo**mentum, a fused optimizer and data parallel algorithm (DeMo) that reduces the communication requirements by several orders of magnitude, potentially enabling future training of large neural networks on slow internet bandwidths with heterogeneous networking hardware. Furthermore, our method is agnostic to the network topology and neural network architecture, and supports scalable clock-synchronous distributed training with negligible compute and memory overhead. Empirically, we show that models trained with DeMo match or surpass the performance of equal models trained with AdamW, entirely bypassing the need for high-speed interconnects for pre-training large scale foundation models.

## 1 INTRODUCTION

Large scale neural networks, particularly language models, are characterized by high parameter counts. Indeed, it is not uncommon to talk about models with trillions of parameters. To train these models, multiple accelerators (e.g. GPUs, TPUs) must be used to make training time tractable. There are multiple strategies to split the training among these accelerators, such as Distributed Data Parallelism (Li et al., 2020) and Fully Sharded Data Parallelism (Zhao et al., 2023). These techniques work by having the accelerators split the weights and synchronize the gradients (sometimes multiple times per step), the size of which is on the order of the size of the model itself.

Gradient synchronization between accelerators necessitates the use of specialized high-speed interconnects (e.g. Infiniband). These interconnects are expensive localized networking topologies, which requires that all accelerators be present in the same data center. However, if the amount of data needed to be synchronized could be massively reduced, these requirements could be relaxed.

In this paper we show that the gradients and optimizer states during training of many large scale neural networks are redundant and highly compressible. Armed with this knowledge, we develop DeMo, an optimizer that leverages this compressibility to reduce the communication needs between accelerators by several order of magnitudes. To evaluate DeMo, we train a standard LLM architecture (decoder-only Transformer (Vaswani, 2017)) using the baseline optimizer (AdamW (Loshchilov & Hutter, 2019)) on traditional high-speed interconnect as well as with DeMo under bandwidth constrained scenarios, and show that the models trained with DeMo meet or exceed the performance of their vanilla counterparts.

## 2 BACKGROUND AND RELATED WORK

To mitigate the communication overhead in distributed training, a variety of strategies have been employed over the years. The most effective techniques for centralized and clock-synchronous training can be sorted into three broad categories:

- Quantization and sparsification of gradients.

- Low rank projection of gradients.

- Federated averaging (also known as Local-SGD).

It is important to note that this work will not consider asynchronous methods, decentralized methods or any method that requires the use of specific network topologies or neural network architectures. Such methods constitute a distinct class of techniques; they introduce considerable complexity in analysis, are not generalizable to all use cases, and involve numerous factors that are beyond the scope of this study. Rather, we will restrict our focus to the simplest and most generalizable scenario: a centralized and clock-synchronous distributed optimizer.

## 2.1 QUANTIZATION AND SPARSIFICATION

Previous work, such as (Wang et al., 2023), focuses mainly on the compressibility of the gradients as-is, within the assumption that the gradient values are uncorrelated and tolerant to quantization and sparsification errors. Of note, the compression gain that can be obtained from quantization is bounded – a 16-bit gradient can only be at most compressed down to one bit. The gain from sparsification is unbounded, but sparsifiction can only achieve limited compression ratios without hampering training, and is therefore best suited for fine-tuning existing models.

## 2.2 LOW RANK PROJECTION

It was shown in (Zhao et al., 2024) that the gradients of LLMs are very low rank during training. As such, a Singular Value Decomposition (SVD) can be used to find a low rank projection matrix that preserves the most significant directions of the gradient, and the model can be optimized using only the projected gradients. This drastically reduces the size of the gradients and optimizer states that have to be stored and transmitted to other accelerator nodes. A disadvantage of this approach is that finding the SVD of the gradients of a very large model is expensive, and the relatively large projection matrix must be shared or recomputed across all nodes. This overhead can be reduced by not recomputing the SVD and projection matrices every single step, but it remains a significant bottleneck that only gets worse as the model size increases. However, as this method achieves convergence[1] parity with full rank optimizers when pre-training LLMs, it teaches us a very important and useful lesson: that compression using a low rank projection is better than sparsity and should be investigated further.

## 2.3 FEDERATED AVERAGING

In (McMahan et al., 2017), federated averaging is used to reduce the amount of communication needed between nodes during a distributed training run. Essentially, each accelerator node trains independently for a fixed number of steps, then synchronizes the accelerator nodes by averaging their weights, which means that the gradient and optimizer states do not need to be communicated for every single step. However, at synchronization this method still requires sharing the full parameter weights across every node, which has the same order of bandwidth cost as unmodified training. Furthermore, increasing the number of steps between synchronizations hurts convergence, exchanging one training speed bottleneck with another. In one extreme, the training iterations run very fast because there is little communication between the nodes, but convergence is very slow. In the other extreme, we can have convergence parity with traditional optimizers, but the time spent synchronizing the parameters becomes significant, so on average one iteration will be prohibitively slow. The optimal point is somewhere in the middle. In practice, federated averaging slows down training by a non-negligible factor and does not scale well with respect to the number of accelerator nodes, which makes it difficult to find the optimal hyper-parameters[2] for specific training runs.

---

[1] Rate of loss decrease per iteration, or per minibatch of data.

[2] A lot of extraneous variables unrelated to the optimizer itself will affect the optimal hyper-parameters, such as (but not exclusively), the number of accelerator nodes, the network bandwidth, the neural network architecture, the batch sizes, etc. In most cases, federated averaging only works on a case-by-case basis, and is not suitable as a drop-in optimizer replacement.

## 3 METHODOLOGY

Rather than relying on ad-hoc modifications of existing optimization algorithms, we propose in this work a novel and general decoupled momentum optimization algorithm that allows and utilizes different optimizer states between accelerators.

### 3.1 ASSUMPTIONS

To formulate our method, we made three crucial assumptions that, while currently lacking theoretical proof, show indications of validity based on empirical evidence.

**Conjecture 3.1** *Fast moving components of the momentum are highly spatially auto-correlated, most of the energy of the momentum is concentrated in a few principal components.*

**Conjecture 3.2** *Fast moving components of the momentum have low temporal variance and should be used to update the parameters immediately, whereas slow moving components of the momentum have high temporal variance and should be smoothed out temporally and be used over a longer period of time.*

**Conjecture 3.3** *Slow moving components of the momentum are very important for long-term convergence, they should not be filtered out or be discarded.*

We will not formally prove any of these conjectures in this work, but the optimizer that we show was made with all of these assumptions in mind. We hope that by proposing this novel method, it can help develop these ideas further in future research.

### 3.2 ALGORITHM

Starting from the SGD with Momentum optimization algorithm, we first remove the all-reduce operation on the gradients $\tilde{g}_k$, decoupling the momentum $m$ in each accelerator node. Then, after updating the momentum, we extract and remove from it the fast components $q$, which can be synchronized very efficiently with a minimal amounts of data being transmitted. The pseudo-code of the algorithm is described in **Algorithm 1**.

---
**Algorithm 1** Decoupled Momentum Optimization

---
**Input:** learning rate $\eta$, decay $\beta \in (0,1)$, parameters $x_t$, momentum $m_t$, hyperparameters $s, k$
$\tilde{g}_t \leftarrow \text{LocalStochasticGradient}(x_t)$       $\triangleright$ Get local gradient $g$ without all-reduce
$m_t \leftarrow \beta m_t + \tilde{g}_t$           $\triangleright$ Accumulate gradient in momentum $m$
$q_t \leftarrow \text{ExtractFastComponents}(m_t, s, k)$     $\triangleright$ Extract fast components $q$ from $m$
$m_{t+1} \leftarrow m_t - q_t$             $\triangleright$ Remove $q$ from $m$
$Q_t \leftarrow \text{Synchronize}(q_t)$       $\triangleright$ Synchronize $q$ across all accelerators
$x_{t+1} \leftarrow x_t - \eta Q_t$            $\triangleright$ Parameter update step

---

#### 3.2.1 EFFICIENT EXTRACTION OF FAST MOVING COMPONENTS

In order for our method to work, we first have to decorrelate, separate and extract the principal components from the momentum during training. Assuming 3.1 is true, one way would be to apply a spatial Kosambi–Karhunen–Loève Transform (KLT) to the momentum in order to separate the faster moving components from the slower ones. However, computing the KLT on the momentum of a large neural network with billions or even trillions of parameters is prohibitively expensive.

Alternatively, taking cues from signal processing work, the Discrete Cosine Transform (DCT) can act as an approximation of the KLT, if used for the purpose of energy compaction, as they are both decorrelating transforms. It is shown that for highly spatially correlated signals, the DCT approximation approaches the KLT (Roma & Sousa, 2011). While the DCT approximation is not perfect, it has many advantages that cannot be overlooked. Firstly, the DCT is highly parallelizable and is extremely fast to compute on modern GPUs. Furthermore, it being a separable transform ensures that its computational complexity scales linearly for 2D, 3D or even n-dimensional signals.

Finally, the DCT has a fixed orthogonal basis, which means it is possible to perfectly decode a DCT-encoded signal without any auxiliary information; the transform matrix and its inverse are known in advance.

If 3.1 is true, we can assume that the DCT would be a good approximation to the KLT for extracting the fast moving components from our momentum, as the momentum is assumed to be spatially auto-correlated. Empirically, we found that using the DCT alone is enough to extract a very good approximation of the principal components.

During training, we treat each momentum tensor as a $d$-dimensional auto-correlated signal, and we chunk each momentum tensor of shape $(n_0, n_1, ..., n_{d-1})$ into contiguous chunks with shape $(s_0, s_1, ..., s_{d-1})$, where each $s_i$ is a divisor of $n_i$, and apply a separable $d$-dimensional decorrelating DCT transform on all chunks. Next, we find the top-$k$ DCT frequencies with the biggest amplitudes in each chunk and treat them as if they were the principal components of the momentum. Both $(s_0, s_1, ..., s_{d-1})$ and $k$ are hyperparameters and control the size of the effective "rank" of the frequencies that we extract.

After extracting the highest energy frequencies, we are left with two tensors of size $(\frac{n_0}{s_0}, \frac{n_1}{s_1}, ..., \frac{n_{d-1}}{s_{d-1}}, k)$, one tensor representing the discrete frequency bins as integer indices, the other representing the amplitudes as a floating point number.

Effectively what we have done here is create a fast transform $p$ that tries to maximize "energy compaction". This way, most of the "movement" described by the momentum can be compressed down to fewer numbers without resorting to sparsity or quantization, which is a similar idea to but is not exactly the same as a low rank projection. For ease of reference, we will define the transform $p$ here as follows, where $\boldsymbol{s}$ is the vector representing the chunk sizes for each dimension of $m_t$:

$$\tilde{m}_t^{freq}, \tilde{m}_t^{ampl} = p(m_t, \boldsymbol{s}, k) \tag{1}$$

We can reverse this transform by first scattering both frequency and amplitude tensors onto a sparse tensor chunked the same way as before, then apply the inverse DCT transform, obtaining something close to the "fast moving components" $q$ of the original momentum:

$$q_t = p^{-1}(\tilde{m}_t^{freq}, \tilde{m}_t^{ampl}) \tag{2}$$

The next step's momentum is then set to be equal to the residual, which represents the "slow moving components" of the original momentum:

$$m_{t+1} = m_t - q_t \tag{3}$$

Note that because the principal components are removed from the momentum at each step, the momentum decay rate should be lowered in general. For example, $\beta = 0.999$ would be a more reasonable value for pre-training a LLM, instead of the usual $\beta = 0.9$.

Also, since the DCT is computed on relatively small static chunks of shape $(s_0, s_1, ..., s_{d-1})$, the required transition matrices can be pre-computed in advance and reused at each iteration, which makes the memory and computational overhead almost negligible if implemented correctly.

### 3.2.2 LOW BANDWIDTH SYNCHRONIZATION

After extracting $\tilde{m}_t^{freq}, \tilde{m}_t^{ampl}$ from the momentum $m_t$, we can then perform an all-gather on the last dimension of the extracted bins. This allows us to perform the same inverse DCT operation by scattering both frequency and amplitude tensors the same way as before, but this time we average the amplitude of any duplicate frequencies. If $s$ and $k$ are chosen appropriately, the two tensors $\tilde{m}_t^{freq}, \tilde{m}_t^{ampl}$ can be orders of magnitude smaller than the size of the model, which allows for a communications efficient way of synchronizing the update step across all accelerators.

Given 3.2 and 3.3, here we are effectively averaging all of the fast moving components of the momentum at each step, while letting the slow moving components be decoupled from each-other. If we assume that slow moving components in the gradient are high variance, they will be accumulated over time in the momentum. As such, slow moving components can slowly overtake fast moving components in strength, which is then transmitted and removed from the momentum. From this,

we can conclude that slow moving components are gradually transmitted alongside the immediate transmission of fast components.

Finally, the gradient descent step is described here, where $\eta$ is the learning rate and $Q_t$ the fast moving components of the momentum accumulated from all accelerators:

$$\theta_{t+1} = \theta_t - \eta Q_t \tag{4}$$

### 3.3 SIGNUM

In order to improve convergence when training LLMs, a signum (Bernstein et al., 2018) variant of DeMo can be used instead, where the gradient descent step is replaced by:

$$\theta_{t+1} = \theta_t - \eta \, \text{sign}(\eta Q_t) \tag{5}$$

Since the second moment is not computed here, this variant of DeMo uses less memory for optimizer states as compared to AdamW.

## 4 EXPERIMENTAL RESULTS

We performed experiments on the signum variant of DeMo using OLMo (Groeneveld et al., 2024), a highly reproducible large language model pre-training framework. Adapting OLMo to use DeMo consisted only of including the DeMo optimizer class in the training code. The only additional change needed was disabling the gradient synchronization in PyTorch Distributed Data Parallelism (Li et al., 2020). We have publicly released the modified OLMo code as well as the configuration files for all experiments.

Our experiments used the Dolma v1.5[3] dataset for pre-training. As a baseline we used the publicly released OLMo-1B[4], a standard decoder-only Transformer model consisting of 1.18 billion parameters using the AdamW optimizer ($\beta_1 = 0.9$, $\beta_2 = 0.95$, weight decay $= 0.1$) as compared to using the DeMo optimizer ($\beta = 0.999$).

Due to constrained compute availability we trained models for 100 billion total tokens rather than the full 3 trillion tokens in Dolma. As a secondary baseline and for complete comparability we re-trained OLMo-1B with these same 100 billion tokens and the learning rate schedule adjusted accordingly. We also repeated the experiments on a smaller 300M model identical to the 1B except halving the model's hidden size. All experiments were performed on 64 H100 GPUs with a global batch size of 2048 with a sequence length 2048 tokens, resulting in a per-GPU batch size of 8.

Figure 1 shows the cross-entropy training loss of DeMo to the reference AdamW model for various values of the hyperparameters $k$ and fixed shape[5] of $s = 64$. Additionally we report the final training loss, per-GPU communication requirements, and downstream evaluation scores of the Hellaswag (Zellers et al., 2019), ARC-Easy (Clark et al., 2018), and PiQA (Bisk et al., 2020) tasks for these configurations as well as $s = 128$ in Table 1.

## 5 CONCLUSION

In conclusion, we have shown that our proposed DeMo optimization algorithm can act as a drop-in replacement to AdamW when training LLMs, with no noticeable slowdown in convergence while reducing communication requirements by several orders of magnitude. The signum variant of DeMo is more memory efficient than AdamW and has negligible compute overhead if we use small precomputed DCT transition matrices. Finally, the LLMs pre-trained with DeMo have equivalent or better scores on multiple standard benchmarks compared to their equivalents trained with AdamW.

---

[3]`https://huggingface.co/datasets/allenai/dolma/blob/main/urls/v1_5.txt`
[4]`https://huggingface.co/allenai/OLMo-1B`
[5]For brevity, $s = 64$ means a shape of $(64, 64)$ for a 2D parameter tensor and $(64, ..., 64)$ for a n-D tensor.

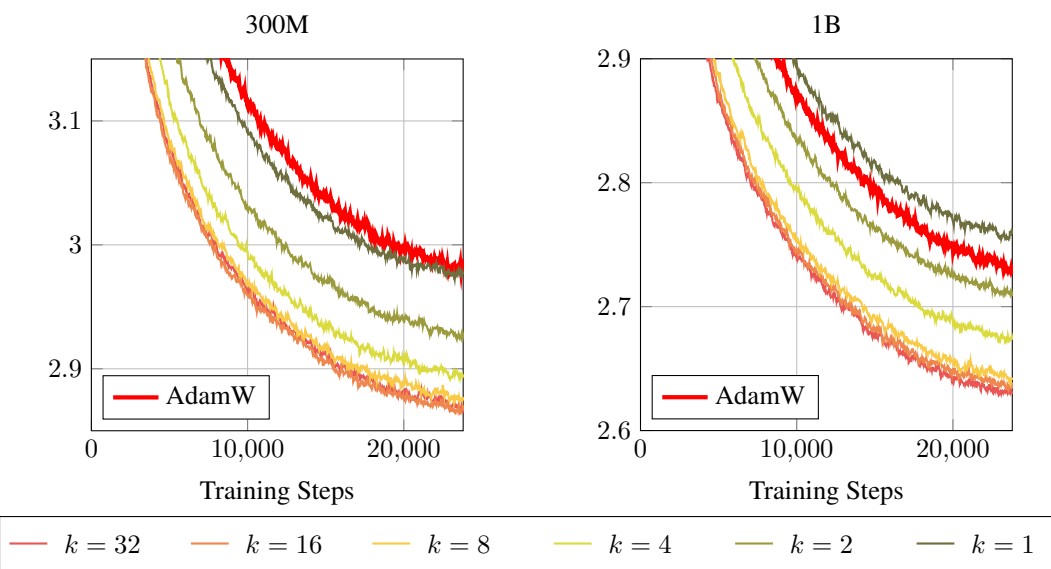

Figure 1: Convergence of training cross-entropy loss across model sizes trained on 100B tokens of reference AdamW and DeMo with $s = 64$ for various values of $k$ hyperparameter

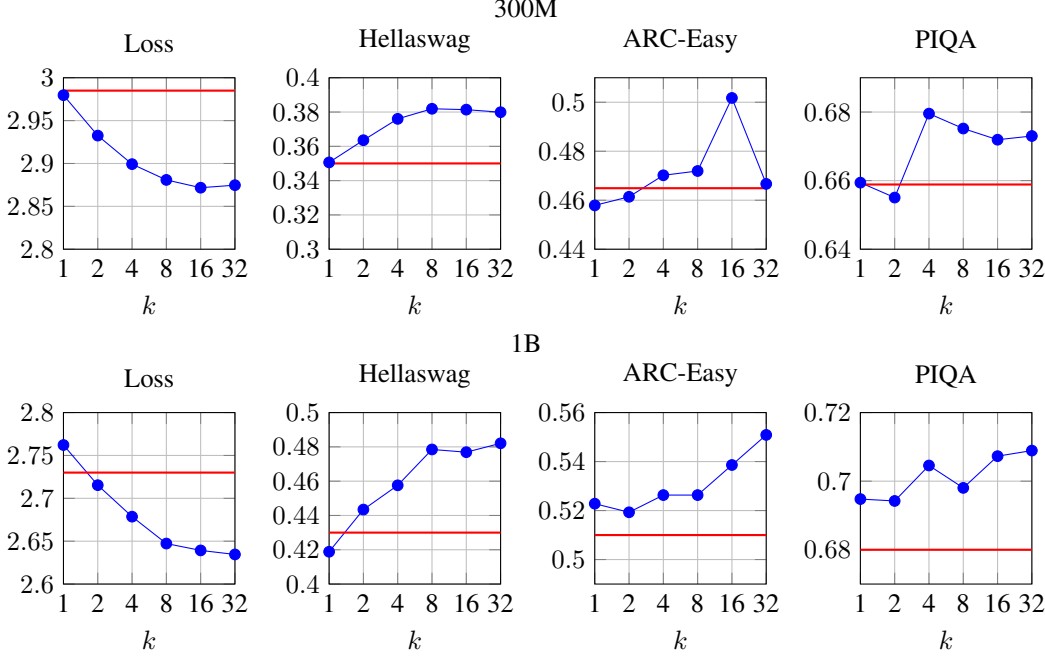

Figure 2: Final loss and downstream evaluation scores of model sizes trained on 100B tokens for various values of the $k$ hyperparameter. The red line represents the reference AdamW training run.

## 6 REPRODUCIBILITY STATEMENT

As described in Section 4 we chose the OLMo framework and references to maximize reproducability and comparability of our experiments. We have provided as publicly available supplementary material a standalone PyTorch implementation of DeMo, as well as the minimal patch to OLMo

| Model | Final Loss ↓ | Hellaswag ↑ acc_norm | ARC-Easy ↑ acc | PIQA ↑ acc_norm | Data Tx ↓ MB/step |
|---|---|---|---|---|---|
| **DeMo 300M** | | | | | |
| $s = 64,\ \ k = 32$ | 2.87 | 0.37 | 0.46 | 0.67 | 29.9 |
| $s = 64,\ \ k = 16$ | 2.87 | 0.38 | 0.50 | 0.67 | 14.9 |
| $s = 64,\ \ k = 8$ | 2.88 | 0.38 | 0.47 | 0.67 | 7.49 |
| $s = 64,\ \ k = 4$ | 2.89 | 0.37 | 0.47 | 0.67 | 3.74 |
| $s = 64,\ \ k = 2$ | 2.93 | 0.36 | 0.46 | 0.65 | 1.87 |
| $s = 64,\ \ k = 1$ | 2.97 | 0.35 | 0.45 | 0.65 | 0.93 |
| $s = 128, k = 32$ | 2.88 | 0.37 | 0.50 | 0.66 | 7.49 |
| $s = 128, k = 16$ | 2.90 | 0.37 | 0.47 | 0.67 | 3.74 |
| $s = 128, k = 8$ | 2.93 | 0.36 | 0.49 | 0.66 | 1.87 |
| $s = 128, k = 4$ | 2.98 | 0.35 | 0.46 | 0.64 | 0.93 |
| $s = 128, k = 2$ | 3.06 | 0.33 | 0.45 | 0.65 | 0.46 |
| $s = 128, k = 1$ | 3.16 | 0.31 | 0.45 | 0.63 | 0.23 |
| **AdamW-DDP 300M** | 2.98 | 0.35 | 0.46 | 0.65 | 636.9 |
| **DeMo 1B** | | | | | |
| $s = 64,\ \ k = 32$ | 2.63 | 0.48 | 0.55 | 0.70 | 110.32 |
| $s = 64,\ \ k = 16$ | 2.63 | 0.47 | 0.53 | 0.70 | 55.16 |
| $s = 64,\ \ k = 8$ | 2.64 | 0.47 | 0.52 | 0.69 | 27.58 |
| $s = 64,\ \ k = 4$ | 2.67 | 0.45 | 0.52 | 0.70 | 13.79 |
| $s = 64,\ \ k = 2$ | 2.71 | 0.44 | 0.51 | 0.69 | 6.89 |
| $s = 64,\ \ k = 1$ | 2.76 | 0.41 | 0.52 | 0.69 | 3.44 |
| $s = 128, k = 32$ | 2.65 | 0.46 | 0.53 | 0.69 | 27.58 |
| $s = 128, k = 16$ | 2.67 | 0.46 | 0.50 | 0.70 | 13.79 |
| $s = 128, k = 8$ | 2.72 | 0.44 | 0.52 | 0.68 | 6.89 |
| $s = 128, k = 4$ | 2.76 | 0.41 | 0.50 | 0.67 | 3.44 |
| **AdamW-DDP 1B** | 2.73 | 0.43 | 0.51 | 0.68 | 2416.6 |

Table 1: Results of training loss, downstream evaluation scores, and per-GPU communication requirements of the model sizes and reference trained on 100B tokens

and configuration files used for the experiments. We do this in hopes of encouraging independent reproduction and improvement of our method.

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
