# OpenReview forum: "DeMo: Decoupled Momentum Optimization"
_ICLR.cc/2025/Conference — ICLR 2025 Conference Withdrawn Submission_

### Official Review · Reviewer_ujxZ · 2024-11-03

**Soundness:** 2
**Presentation:** 2
**Contribution:** 2
**Rating:** 3
**Confidence:** 4

**Summary:**

This paper proposes a new optimizer, *DeMo*, that reduces communication costs in training large-scale neural networks in distributed systems. Unlike commonly used optimizes like SGD or Adam, *DeMo* does not synchronize full gradients when training a neural network in a distributed system. Instead, it first extracts several principal components from the original gradients and then only synchronizes these principal components. Therefore, the communication cost for gradient aggregation is reduced.

The authors adopt the discrete cosine transform (DCT) to extract top-$k$ frequencies as the principal components, and synchronize these DCT coefficients in the distributed system.

The authors compare *DeMo* against AdamW in training a Transformer model, and show that *DeMo* achieves a lower loss and faster convergence.

**Strengths:**

Overall, I think the authors present an interesting idea for reducing communication costs in training large-scale neural networks. In general, the idea that identifies principal components in gradients and only synchronizes the principal part sounds like a valid approach.

Further, the authors adopt an efficient approach based on discrete cosine transform (DCT) to extract the principal components. In distributed systems, DCT can be easily parallelized.

**Weaknesses:**

While I like the idea, I still think the current version of this work has several limitations:

1. **The authors build the idea based on conjectures that need further justification.**

    In Sec 3.1, the authors state three assumptions but without providing supporting analysis or evidence. And the new optimizer is developed based on these conjectures. It is questionable whether the optimizer applies to general cases on large-scale model training. Therefore, I believe the authors at least need to provide some relevant observations or a toy optimization problem to justify these arguments.

2. **Technical terms are ambiguous and not explained.**

    The authors frequently mention *fast and slow moving components* without providing a proper definition or explanation. It is hard to map these terms to an actual optimization scenario. Specifically, when training a large-scale model, it is unclear which part of the gradients can be seen as *fast moving components*.

    Without a proper definition of *fast and slow moving components*, it is not convincing why DCT can be used here to extract fast moving components.

3. **Results are confusing.**

    In the experiments, the authors adjust the parameter $k$ to extract different numbers of principal components. With increasing $k$, the loss becomes smaller. However, it is confusing why *DeMo* with a larger $k$ achieves a lower loss compared to AdamW.

    As shown in Algorithm 1, $k$ controls how much information will be synchronized. An extreme case is that $k$ is large enough so that all gradients are synchronized like in the standard SGD. However, this means the performance of *DeMo* at most can be as good as the standard SGD. It is confusing why it outperforms AdamW, which usually gives much better performance than SGD.

**Questions:**

See the comments above.

---

### Official Review · Reviewer_99ZZ · 2024-11-04

**Soundness:** 1
**Presentation:** 3
**Contribution:** 1
**Rating:** 3
**Confidence:** 3

**Summary:**

This paper introduces DeMo, a novel fused optimizer designed to significantly reduce the communication volume between GPUs during distributed training, thereby accelerating the overall process.  Experimental results demonstrate that models trained with DeMo not only reduce communication overhead but also match or exceed the performance of models trained with the widely-used AdamW optimizer.

**Strengths:**

1. The paper introduces a novel fused optimizer designed to significantly reduce the communication volume between GPUs during distributed training.

2. The experiments conducted with a reproducible large language model (OLMo framework) support the claims that DeMo can match or even exceed the performance of traditional AdamW-based methods.

**Weaknesses:**

1. This paper's design choices are largely heuristic, lacking a strong theoretical basis and formal convergence bound proofs for the DeMo.

2. The Method section of this paper needs to be rewritten to improve clarity and logical flow, making it easier for readers to understand. Additionally, the paper should include more detailed explanations of relevant methods, such as KLT and DCT.

3. In the experimental section, a more in-depth analysis and heuristic guidance on how to select the hyperparameter k should be included, as it is a crucial factor. Additionally, since DeMo demonstrates better convergence than AdamW, further analysis is needed to explain why it outperforms AdamW.

**Questions:**

1. In the supplementary materials, I noticed that the weight-decay settings for the AdamW and DeMo experiments are different, set to 0.1 and 0, respectively. Why was this choice made? As far as I know, setting weight-decay to 0 can lead to faster loss reduction but also increases the risk of overfitting.

---

### Official Review · Reviewer_fsqz · 2024-11-04

**Soundness:** 2
**Presentation:** 1
**Contribution:** 2
**Rating:** 3
**Confidence:** 4

**Summary:**

The paper introduces a new way of compressing the optimizer states, in particular, a momentum using Discrete Cosine Transform, a technique that was used mostly in image compression before. This technique can help to significantly reduce the communication time in Distributed Data Parallel algorithm widely used to train large neural networks.

**Strengths:**

As was discussed a lot in prior literature, e.g. [1], optimizer states represent a heavy memory load that can also impact the communication time during data parallel training. The idea of applying Discrete Cosine Transform to momentum compression and thus reducing the communication load seems to be novel. The Table 1 suggests that this method allows us to lower a lot the load on the communication bus (from 636.9 to 0.23).



1. Rajbhandari, Samyam, et al. "Zero: Memory optimizations toward training trillion parameter models." SC20: International Conference for High Performance Computing, Networking, Storage and Analysis. IEEE, 2020.

**Weaknesses:**

The paper is badly written.

First of all, there is a severe lack of related literature. For each alternative method such as quantization, sparcification, low rank decomposition and federated averaging, only one work per method is cited, while there exists a huge amount of literature, even if we consider quantization alone. The authors have also missed another line of work that relies on gradient compression using learning, see [1].

While the authors consider only Distributed Data Parallel setting, there exists many other parallelism strategies such as pipelined model parallelism, tensor parallelism and hybrid schemes. Depending on the choice of parallelization, the communication load can differ a lot, for example in pipelined model parallelism only activations are communicated which may be much lighter than weights or optimizer states depending on the problem. Even when considering DDP setting alone, the communication load may vary a lot depending on what is communicated, weights or optimizer states, and what bit representation is used. Typically, DDP is combined with mixed precision where weights are stored in a lower-bit format (16-bit usually) while optimizer states in a higher-bit format (32-bit), and in this case it is more practical to do updates locally and after communicate weights only.

All those aspects were almost not discussed, while it is essential to provide a full context to a reader, especially considering that only 7 out of 10 pages were used.

Furthermore, the algorithm proposed in the paper modifies the training dynamics, thus it is crucial to demonstrate the convergence of the new algorithm. There is no theoretical analysis of the convergence and only few experimental results are present in the paper, which is not enough for validation. While DeMO is motivated by three conjectures about training dynamics, these conjectures are just stated and there is no discussion behind that would justify them. Even if proving them is challenging, providing an empirical demonstration should be doable at least for some examples.

Finally, I have also some concerns regarding the experimental results. First of all, there is no comparison with other alternative methods like Galore, while the authors mentioned a few times in a paper that their method should be close to low-rank approaches. Moreover, it is not clear how many iterations were used to train the models. Figure 1 shows that at least 20000 training steps were used, but on Figure 1 we can see that none of the runs reached a convergence, so it is possible that training longer could have changed the final results.

**Remark concerning supplementary materials:**

There is no train.py file, despite that README.md suggests launching it to reproduce the results.

1.Abrahamyan, Lusine, et al. "Learned gradient compression for distributed deep learning." IEEE Transactions on Neural Networks and Learning Systems 33.12 (2021): 7330-7344.

**Questions:**

What is communication bandwidth of your hardware?
Can you please report final training time for each method?

---

### Official Review · Reviewer_7zsu · 2024-11-04

**Soundness:** 1
**Presentation:** 3
**Contribution:** 2
**Rating:** 3
**Confidence:** 2

**Summary:**

This work describes a novel optimization routine that reduces the communication overhead associated with momentum-based DNN training methods. The key idea is that during training, the optimizer and gradient states contain highly redundant information, both spacially and temporally. Unlike related methods, such as those that exploit sparsification and quantization, the technique presented in the paper uses a DCT (discrete cosine transform) to decouple the "fast moving components" of the momentum from the slow moving components and reduce the overhead of communication during training by communicating only the slow moving components. The training results using several different transform sizes demonstrate that the method performs admirably compared to generic AdamW on the same network and training data.

**Strengths:**

- The implementation of the method is described enough. Algorithm 1 clearly outlines the high-level computation of the optimizer updates and the transform computation is discuss in detail in Section 3.2.1.
- The overall writing is not overly long and verbose and clearly states the authors limitations with respect to proving the conjectures proposed in Section 3.
- The results in the experimental section support the authors claims that the method effectively replaces AdamW for the test case they investigated.
- The idea seems uniquely novel to me and the author differentiate DeMo compared to other methods that rely on sparsification, quantization, and/or averaging.

**Weaknesses:**

- Given the lack of proof for the conjectures proposed in Section 3 I think a more extensive evaluation of the method must be presented to support the empirical results. If the method reduces communication during training I would have liked a plot to show the reduction in communication as the dimension, I believe $k$, is changed from 1 to 64 or at least the impact of DeMo on overall training time. Though the proposed method appears promising and novel a more thorough presentation in the evaluation section is required to verify the merits.
- Limited evaluation on a single DNN and dataset with little motivation makes it difficult to properly evaluate the effectiveness of the changes in different training scenarios.
- This point ties into the previous comment but 3 conjectures were proposed with no motivation or proof which, combined, forms a large barrier for the reader to agree with for the results to seem reasonable.
- The signum variant in Section 3.3 seems tacked onto the optimizer but no comparisons are made or issues with the vanilla version are explicitly noted.
- Aesthetically, Figure 1 was hard on my eyes, I would recommend using dashed lines and symbols to make it easier to differentiate the plots.
- Based on these observations I would recommend a few more iterations to improve the quality of the presentation focusing on the evaluation section.

**Questions:**

- If training overall is the primary constraint could you supply training results on other models, such as vision models? Though distributed communication is not necessary in those cases the ability to run a large corpus of tests and collect training data quickly for many architectures may be inciteful.
- Does the method work but not as well without the signum variation?
- Is the terminology "slow-moving" and "fast-moving" components common when discussing momentum-based optimization? I'm familiar with high-frequency and low-frequency or high energy and low-energy but fast and slow are somewhat new for me and I want to make sure I understand the component of the error they are referring to.

---

### Official Review · Reviewer_PuUt · 2024-11-05

**Soundness:** 1
**Presentation:** 3
**Contribution:** 1
**Rating:** 1
**Confidence:** 4

**Summary:**

The paper proposes decoupled momentum optimization (DeMo), an approach to significantly reduce the communication volume (hence, bandwidth) needed when during training by compressing the momentum before communication. The compression is based on the idea that there is significant redundancy in the momentum and only specific components need to be communicated in each iteration. An empirical study using the OLMo-1B model shows improved convergence over AdamW.

**Strengths:**

1. The paper aims to reduce communication volume when training, which helps to lower the barrier to training or fine-tuning large models as expensive, high-performance networks are not required.
2. The paper is clearly, identifies its hypotheses and assumptions, and explains its algorithm.
3. The experimental results are at reasonable scale (64 GPUs) and involve a modern LLM and show generally improved convergence.

**Weaknesses:**

1. While the paper defines its assumptions (Conjectures 3.1-3.3), these are never even validated empirically. While it may be okay that they are not theoretically justified, the paper lacks any justification at all for these. Can they be demonstrated experimentally?
2. The experimental comparison does not demonstrate any end-to-end speedup from DeMo. What is the time per iteration and/or time to converge to a fixed loss with DeMo? While many communication compression methods are able to reduce communication volume, this is insufficient for two reasons: (1) communication is often overlapped with computation, making it less of a bottleneck; and (2) the overhead of compression/decompression can outweigh any wall-time benefit from reduced communication volume.
3. A related point: The paper does not include an analysis or breakdown of the runtime of a training iteration. How expensive is the compressor?
4. There are no comparisons to other compressed communication systems. Many of these have been proposed (as the paper discusses) and it is not clear that DeMo outperforms them, either in terms of convergence of wall-time performance. Some suggestions for reasonable comparisons include signSGD (Bernstein et al. 2018, cited in the paper), deep compression (Han et al., "Deep Compression: Compressing Deep Neural Networks with Pruning, Trained Quantization and Huffman Coding", ICLR 2016), or SparCML (Renggli et al., "SparCML: High-Performance Sparse Communication for Machine Learning", Supercomputing 2019). How does DeMo compare?
5. While the results with 64 GPUs are valuable, the paper does not demonstrate the effect of the number of processors used on quality. How does DeMo scale?
6. The generalizability of DeMo is not demonstrated. Is it only useful for LLMs, or can it be applied to other model architectures or tasks (e.g., ViTs, CNNs)?
7. I find this statement in the paper unclear: "... this work will not consider ... decentralized methods". Unfortunately, the literate is somewhat inconsistent on what "decentralized methods" are in this area, so I would appreciate greater clarity here. Does this refer to "allreduce-based" methods that eschew a centralized parameter server, or to gossip (or similar) algorithms?

Minor point: The paper states that "a 16-bit gradient can only be at most compressed down to one bit". I think this is not quite true, unless it refers to compressing a single word at a time. A vector can, in principle, achieve higher compression (e.g., with block/vector-based quantization) by exploiting the correlation across values.

**Questions:**

Please see the comments and questions above under "Weaknesses".

---

### Note · Authors · 2024-12-02

**Comment:**

We want to thank every reviewer for your thoughtful feedback and valuable insights. We have decided to withdraw our paper. We sincerely appreciate your time and effort in reviewing our work.

**Withdrawal Confirmation:**

I have read and agree with the venue's withdrawal policy on behalf of myself and my co-authors.